# Improving the Oxidation Resistance of G115 Martensitic Heat-Resistant Steel by Surface Treatment with Shot Peening

Pengwen Chen [1], Jingwen Zhang [1,*], Liming Yu [1,2,*], Tianyu Du [2,3,*], Huijun Li [1], Chenxi Liu [1], Yongchang Liu [1,2], Yuehua Liu [2,3] and Baoxin Du [2,3]

1   School of Materials Science and Engineering, Tianjin University, Tianjin 300350, China; ycliu@tju.edu.cn (Y.L.)
2   Xinyu Joint Research Center, Tianjin University, Tianjin 300072, China
3   Tianjin Xinyu Color Plate Co., Ltd., Tianjin 300382, China
*   Correspondence: jwz@tju.edu.cn (J.Z.); lmyu@tju.edu.cn (L.Y.); tydu@tju.edu.cn (T.D.)

**Abstract:** G115 steel is a novel martensitic heat-resistant steel, primarily utilized in the main steam pipelines and collectors of ultra-supercritical thermal power units. However, the oxidation resistance of martensitic steels in the high-temperature steam environment is usually suboptimal, significantly affecting the efficiency of power plants. In this paper, shot peening (SP) is employed as a surface treatment method for G115 steel, and the oxidation kinetics, oxide layer thickness, and microstructure of shot-peened G115 samples are compared with those of G115 steel. The results indicate that in the 650 °C steam environment, the oxidation kinetics of the shot-peened samples follow the parabolic law and that the oxidation weight gain is significantly smaller than that of the non-shot-peened samples. The higher the SP intensity, the smaller the oxidation weight gain and the better the oxidation resistance. This can be attributed to the fragmentation of the grains in the surface layer caused by external stress during SP, which creates a multitude of grain boundaries that can provide rapid diffusion pathways for corrosion-resistant Cr atoms, resulting in the accelerated outward diffusion of Cr atoms from the substrate. Simultaneously, a continuous and dense $FeCr_2O_4$ protective layer is produced at the interface between the SP layer and the substrate, obstructing the inward diffusion of oxygen and enhancing the oxidation resistance of G115 steel.

**Keywords:** G115 steel; surface treatment; shot peening; oxidation resistance





## 1. Introduction

With the continuous growth in the global electricity demand, thermal power units, as the core source of electricity, have become increasingly significant [1–3]. Within the framework of the current energy structure, enhancing the power generation efficiency of thermal power units is not only a key strategy to meet the rising electricity demand but also a crucial aspect involved in achieving sustainable energy utilization. In recent years, ultra-supercritical thermal power units have progressively replaced previous units, and their share in the thermal power generation sector has considerably increased, which has substantially improved the efficiency of power generation [4–6]. G115 martensitic heat-resistant steel, as a novel material capable of stable operation under higher temperatures and pressures, has now been applied in some critical positions, such as the main steam pipelines of ultra-supercritical thermal power units [7,8]. As one of the core components of ultra-supercritical thermal power units, G115 steel, due to its long-term exposure to high-temperature steam environments, inevitably suffers certain damages. These damages are primarily manifested as the corrosion and peeling of the oxide scale on the inner walls of the pipelines [9–11], not only reducing the power generation efficiency but also posing potential safety risks. Against this backdrop, new requirements for the oxidation resistance of materials at higher operating temperatures have been proposed.

To enhance the service lives of materials in thermal power units and reduce the weight loss caused by the peeling of external oxide scales, numerous methods have been

explored to improve the oxidation resistance of such materials. Zhou et al. [12] employed an aluminizing surface treatment on P92 steel and produced a β-FeAl coating by using a low-temperature method, effectively protecting P92 steel from oxidative degradation in high-temperature steam environments. Wang et al. [13] studied the steam oxidation behavior of chromium coatings prepared on zirconium alloy claddings via magnetron sputtering, and it was found that the coatings could effectively reduce the oxidation rate of zirconium alloy and inhibit the formation of the outer $ZrO_2$ layer, thereby improving the oxidation performance of the material. Liang et al. [14] prepared a corrosion-resistant $(Fe, Cr)_2O_3$ oxide film on the surface of T91 steel in an ozone atmosphere by pre-oxidation treatment under low oxygen pressures, which enhanced the corrosion resistance of T91 steel in the high-temperature steam environment. In the method of adding corrosion-resistant Al and Cr elements to produce antioxidant coatings, or preparing protective oxide films via a pre-oxidation treatment that involves preferentially diffusing Cr to the material surface under low oxygen pressures, the oxidation resistance of the material can be significantly improved. Unfortunately, these methods often cannot be applied in engineering practices for large-scale mass production due to their high costs and poor practicability.

Considering the occurrence of high-temperature oxidation on the material surface, the state of surface grains also significantly affects the steam oxidation behavior [15,16]. From the perspectives of practicality and cost, surface grain refinement can be employed to enhance the oxidation resistance of materials. Valérie Parry [17] investigated the effect of cold working treatments on the oxidation performance of austenitic stainless steels in a high-temperature steam environment. It was found that the recrystallization process induced by cold working could produce numerous fine grains in the early stages of oxidation and provide pathways for rapid chromium diffusion, thus effectively slowing the oxidation process. Gao et al. [18] examined the high-temperature oxidation behavior of annealed and 30% cold-rolled aluminum-forming austenitic (AFA) steels in dry air at 700 °C and discovered that cold rolling affected the oxidation kinetics, phase composition, and microstructural characteristics, leading to grain refinement on the steel surface. In addition, they reported that dislocations could serve as short-range diffusion pathways for aluminum atoms and offered nucleation sites for the B2-NiAl phase, which facilitated the formation of an $Al_2O_3$ film and thus enhanced the overall oxidation resistance.

Shot peening (SP), as a commonly used surface treatment method for metal materials, can significantly improve the oxidation resistance of steel [19–22]. During the SP process, high-velocity projectiles are propelled toward the surfaces of steel tubes, and the workpiece surface layer undergoes plastic deformation and achieves grain refinement due to the impact of the projectiles [23]. SP offers multiple advantages including a desirable cost, simple operation, and convenient use and has already been applied to the service materials of some supercritical thermal power units, significantly extending the service lives of pipelines [24,25]. Some research on the engineering aspects of SP has also been conducted. Kurley et al. [26] and Yue et al. [27] performed SP on TP304H and HR3C austenitic steels, respectively, and it was found that the oxidation resistance of the steels in the high-temperature steam environment could be greatly improved. This demonstrates that the SP process can be utilized in engineering practices to enhance the oxidation resistance of austenitic heat-resistant steels. However, its application on martensitic heat-resistant steels with lower chromium content still requires further study.

This study is dedicated to investigating the changes in the oxidation layer of G115 steel after SP in the 650 °C steam environment through in-depth analysis and experimentation. By comparing the oxidation performance of G115 steel under different SP intensities, the mechanism by which SP enhances the oxidation resistance of G115 steel is explored, and an efficient surface treatment method for G115 steel is determined to achieve desirable stability and durability under extreme conditions. Our work provides solid experimental data and technological support for the enhancement of the reliability and sustainability of thermal power units.

## 2. Materials and Methods

### 2.1. Sample Preparation

The oxidation experiments were conducted by using G115 martensitic heat-resistant steel as the research object, with its chemical composition shown in Table 1. The heat treatment process included normalizing at 1120 °C for 1 h and air cooling to room temperature, subsequent tempering at 780 °C for 3 h, and final air cooling to room temperature. The samples were machined into rectangular plates with dimensions of 10 mm × 8 mm × 2.5 mm and all surfaces were uniformly polished. After the surface oxide layer was removed, the samples were ultrasonically cleaned in acetone for 15 min and then air-dried under a dryer before weighing for use.

**Table 1.** Chemical composition of G115 steel.

| Element | C | Si | Mn | Cr | W | Nb | V | B | Co | N | Cu | Fe |
|---------|-------|------|------|------|------|-------|------|--------|------|-------|------|------|
| % | 0.084 | 0.31 | 0.50 | 8.92 | 2.89 | 0.057 | 0.19 | 0.0092 | 3.02 | 0.018 | 1.09 | Bal. |

During the SP treatment, the surface layer of the sample undergoes plastic deformation due to the impact of the projectiles. The plane serving as the baseline is cut into the deformed spherical surface, and the distance from this baseline to the highest point of the sphere is referred to as the arc height. The SP intensity is denoted by the arc height plus the type of Almen strip, with the arc height measured in millimeters (mm). N, A, and C Almen strips with different thicknesses present different SP effects, and the A strip is the standard for the shot peening intensity (SPI) in experiments. In this study, during SP, all samples were divided into three groups and treated with different SPIs. Each group of samples was uniformly bombarded with stainless steel shots of 0.4 mm in diameter, by using the projectile type of AWC20. The specific parameters for the shot-peened samples are presented in Table 2.

**Table 2.** Experimental parameters of different SPI samples.

| Sample Type | SPI | Shot Peening Time (s) | Surface Coverage |
|-------------|-------|-----------------------|------------------|
| SP0.1 | 0.1A | 60 | 100% |
| SP0.16 | 0.16A | 60 | 100% |
| SP0.23 | 0.23A | 60 | 100% |

### 2.2. Experimental Design

The oxidation experiments were conducted on a high-temperature steam oxidation experimental platform, and a schematic of the platform is illustrated in Figure 1. The platform primarily consisted of a water system, gas system, steam generation system, experimental reaction system, and condensation circulation system. Distilled water, with oxygen content of approximately 30 μg/L, was used as the source of steam for the experiments.

Before the start of the experiment, each group of samples was placed in separate alumina tubular crucibles to suspend the samples in the air, which prevented the bottom of the sample from touching the crucible, thus inhibiting the occurrence of oxidation. During the experiment, the experimental platform was sealed, and argon gas was continuously introduced to maintain a protective environment in the reaction chamber, simultaneously expelling residual oxygen from the reaction furnace. The flow rate of argon was controlled at about 36 mL/min under a flow meter. After the temperature control program was started, the reaction furnace was heated up; when the temperature reached approximately 500 °C, the water circuit valve was opened. A peristaltic pump was used to control the water flow rate (0.36 L/h) to achieve a controlled steam flow, with a steam volume fraction of about 15%. Once the temperature stabilized at 650 °C, the timer was started. The samples were removed from the furnace at set intervals (24 h, 72 h, and 200 h), cooled to room temperature, weighed, and then characterized for analysis.

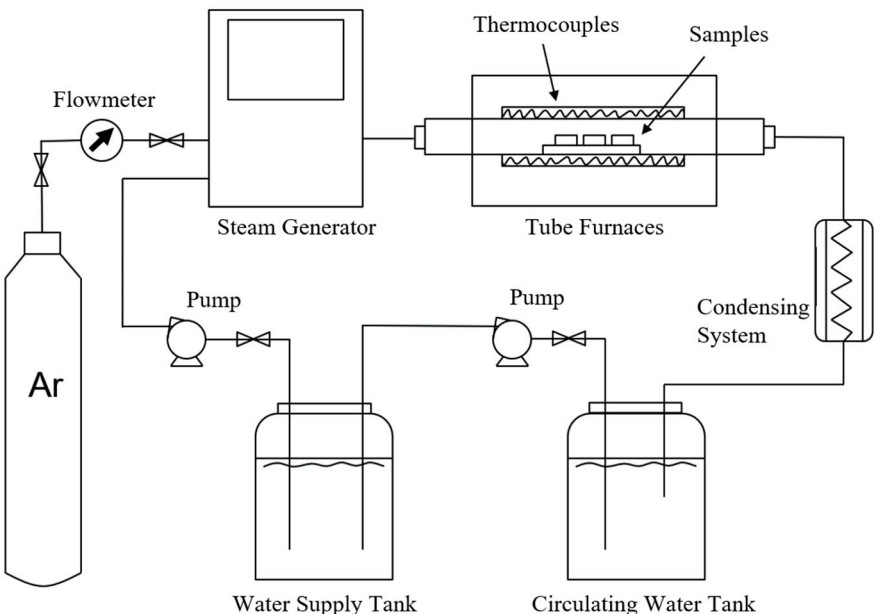

**Figure 1.** Schematic diagram of high-temperature steam oxidation platform.

Upon the completion of all experiments, the oxidation kinetics curves were plotted to observe the time-dependent oxidation weight gains of different samples. The oxidation weight gain per unit area, $\Delta W$, can be calculated according to Equation (1):

$$\Delta W = \frac{m_t - m_0}{S} = \frac{\Delta m(\text{mg})}{S(\text{cm}^2)} \tag{1}$$

where $\Delta m$ represents the change in the weight of the sample before and after the oxidation experiment, and $S$ is the total surface area of the sample.

According to previous studies related to martensitic heat-resistant steels [28,29], it is understood that the oxidation of the samples in this study conformed to the parabolic oxidation law and that the produced oxide film had a protective effect. Fitting the sample oxidation data to the parabolic law yields the oxidation kinetic equation shown in Equation (2), which provides a more intuitive representation of the differences in the oxidation resistance of different samples.

$$\Delta W = kt^n \tag{2}$$

where $k$ is the oxidation rate constant, $t$ is the oxidation time, and $n$ is the time exponent that represents the speed of the oxidation process, and a greater value of $n$ indicates a faster oxidation reaction.

### 2.3. Characterization

The mass of the sample was determined by using a METTLER TOLEDO analytical balance with a resolution of 0.01 mg. The oxidized sample was weighed along with its peeled oxide scales to obtain its total mass after oxidation.

The phase structure of the surface oxides on the samples was identified with a step time of 0.15 s within a 2θ range of 20~90° by using a D8 ADVANCE X-ray diffractometer (XRD) equipped with a Cu target.

The morphology of the surface oxides and the cross-section of the oxidation layers on the specimens were observed using a JSM-7800F scanning electron microscope (SEM) at a working voltage of 15 kV and a current of 200 nA. Before the observation of the cross-sectional morphology of the oxide scale, the samples were embedded in epoxy resin, ground, and polished. The distribution of the chemical composition within the oxide scales was analyzed by using an energy-dispersive spectrometer (EDS).

An FEI Talos F200X G2 transmission electron microscope (TEM) was employed to analyze the oxidation layers and identify the material structure. Before observation, a ZEISS Crossbeam 540 focused ion beam scanning electron microscope (FIB-SEM) was utilized for site-specific preparation, cutting a region of 10 μm × 5 μm for examination.

## 3. Results

### 3.1. Cross-Sectional Morphology of Surface Layer after Shot Peening

Figure 2a–c present the cross-sectional morphologies of the SP layers in the samples at different SPIs. The depth of the SP layer increases progressively with the increase in the SPI. At SP0.16, the depth reaches 25.21 μm, approximately 13.6% greater than that at SP0.1; at SP0.23, the depth is 28.44 μm, about 28.2% greater than that at SP0.1. After SP treatment, the G115 steel exhibits grain refinement within the SP layer, resulting in significantly smaller grain sizes than that within the substrate interior. Notably, different from other SP samples, the grains within the SP layer at SP0.23 are elongated into strip shapes, probably due to the greater external stress caused by the higher SPI.

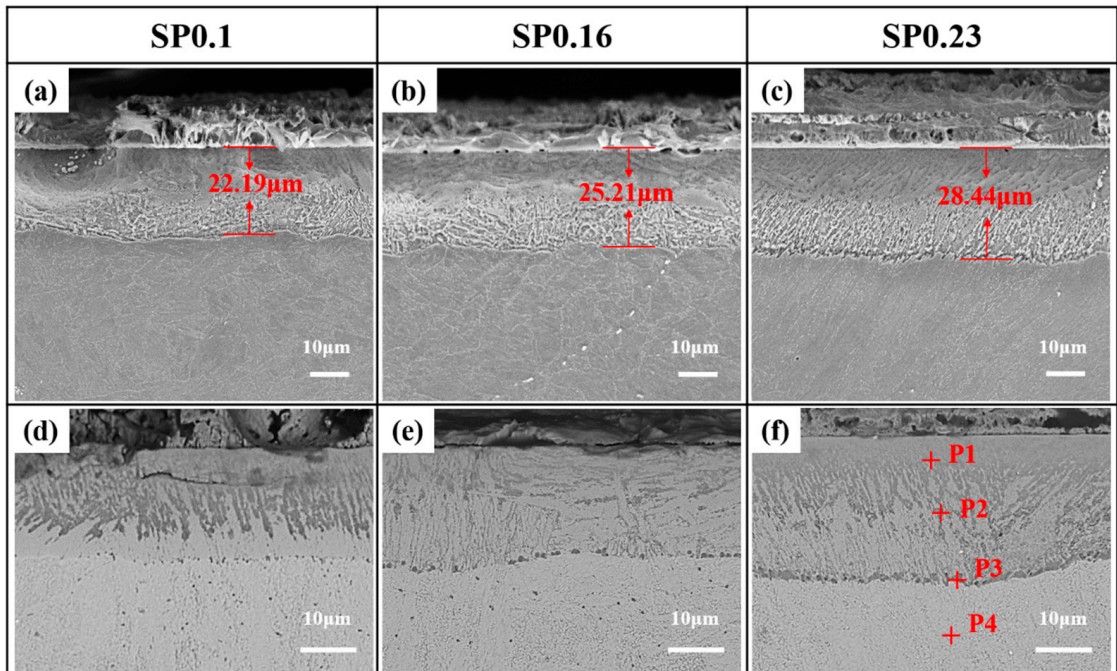

**Figure 2.** SEM cross-sectional morphologies of the samples after shot peening at different SPIs: (**a**) SP0.1, (**b**) SP0.16, and (**c**) SP0.23. BSE images of cross-sectional morphologies: (**d**) SP0.1, (**e**) SP0.16, and (**f**) SP0.23.

Figure 2d–f, presenting the BSE morphology, offer a more intuitionistic observation of the fragmentation and elongation of the surface grains caused by SP treatment. As can be seen, the crushed grains display different orientations. With the increase in the SPI, significant elemental enrichment occurs at the interface between the SP layer and the substrate. A point scanning analysis of the SP layer at SP0.23, as shown in Table 3, reveals that the proportion of Cr within the layer is higher than that in the substrate; the closer to the surface, the lower the proportion of Cr. The outermost layer of SP0.23 still contains 14.62% Cr, exceeding the Cr proportion in the substrate. Therefore, it can be inferred that the SP-induced fragmentation of the grains and the increase in grain boundaries at the surface layer of G115 steel provide numerous diffusion pathways for atoms, facilitating the migration of Cr atoms towards the sample surface, as well as their substantial accumulation at the interface between the SP layer and the substrate.

**Table 3.** EDS point analysis results for marked positions in Figure 2f.

| Position | Element Content (at%) | | | | | |
|---|---|---|---|---|---|---|
| | **O** | **Cr** | **Fe** | **W** | **Co** | **Cu** |
| P1 | 27.24 | 14.62 | 53.6 | 1.34 | 1.07 | 0.29 |
| P2 | 15.48 | 18.19 | 60.54 | 2.14 | 0.15 | 0.22 |
| P3 | 7.15 | 24.81 | 61.23 | 2.03 | 1.27 | 0.46 |
| P4 | 1.04 | 9.14 | 78.69 | 2.97 | 3.22 | 0.91 |

*3.2. Oxidation Kinetics Curves*

Figure 3 shows the oxidation kinetics curves of SP and Un-SP samples oxidized for 200 h in 650 °C steam environment. As can be seen in Figure 3a, when the oxidation time is the same, the Un-SP sample shows a much larger oxidation weight gain compared to the SP samples. As the oxidation proceeds, both SP and Un-SP samples exhibit an increase in oxidation weight gain, kinetically demonstrating parabolic oxidation behavior. By fitting the data points corresponding to the different oxidation times of 24 h, 72 h, and 200 h, the oxidation kinetic equations for each sample can be obtained as follows:

$$\Delta W_{Un-SP} = 0.837t^{0.412} \tag{3}$$

$$\Delta W_{SP0.1} = 0.209t^{0.414} \tag{4}$$

$$\Delta W_{SP0.16} = 0.32t^{0.313} \tag{5}$$

$$\Delta W_{SP0.23} = 0.29t^{0.238} \tag{6}$$

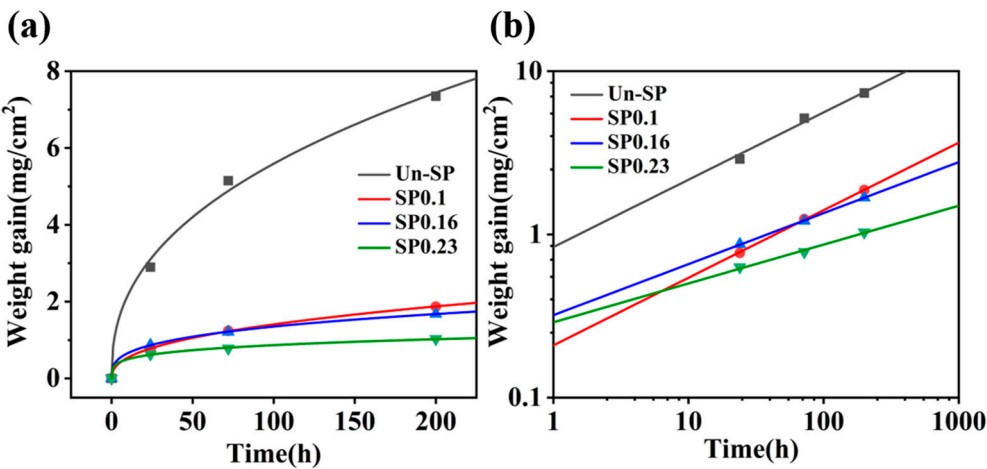

**Figure 3.** (**a**) Oxidation kinetics curves of Un-SP and SP samples after 200 h of steam oxidation at 650 °C, (**b**) logarithmic curves of oxidation kinetics.

Figure 3b presents the oxidation weight gain on a logarithmic scale, where the slope represents the rate of weight gain due to oxidation. On this scale, it is evident that the oxidation weight gain of all samples increases linearly during the oxidation time of 200 h. Furthermore, compared to SP0.1 and SP0.16, the SP0.23 sample exhibits a faster weight gain rate in the initial phase but a slower weight gain rate in the long term. This corresponds to a larger oxidation rate constant $k$ but a smaller time exponent $n$ in the kinetic fitting curve of the SP0.23 sample (Equation (6)). From Equations (4)–(6), it can be seen that as the SPI increases, the value of time exponent $n$ gradually decreases. Accordingly, the samples with a greater SPI show a more smoothly increasing trend in oxidation weight gain, indicating better oxidation resistance.

### 3.3. Surface Morphology and Composition

Figure 4 displays the SEM morphologies of the oxide layers on the surfaces of Un-SP and SP samples after different oxidation times. After oxidation for 24 h, the surface of the Un-SP sample (Figure 4a) shows localized bulges; in contrast, there are no obvious bulges on the surfaces of SP samples (Figure 4d,g,j). This difference is attributed to the stress on the surfaces of the samples induced by high-velocity projectiles during shot peening, producing a compact surface layer of fragmented grains. At SP0.1, the surface oxide layer is flake-like, with no precipitated products; for SP 0.16 and SP0.23, white oxide precipitates appear on the surface. Specifically, at SP0.23, many white oxides are produced, and shot peening-induced pits are observable on the sample surface. This indicates that a higher SPI facilitates the precipitation of oxides from the substrate. After oxidation for 72 h, the surface of the Un-SP sample (Figure 4b) exhibits distinct, flat-faced irregular cubes with typical crystalline geometric shapes. In contrast, the surfaces of the SP samples exhibit sparse holes and are characterized by a combination of bulky cubes and flocculent structures (Figure 4e,h,k). After oxidation for 200 h, both the Un-SP (Figure 4c) and SP samples (Figure 4f,i,l) display fluffy structures on the surfaces, with larger depressions that constitute ravine-like features.

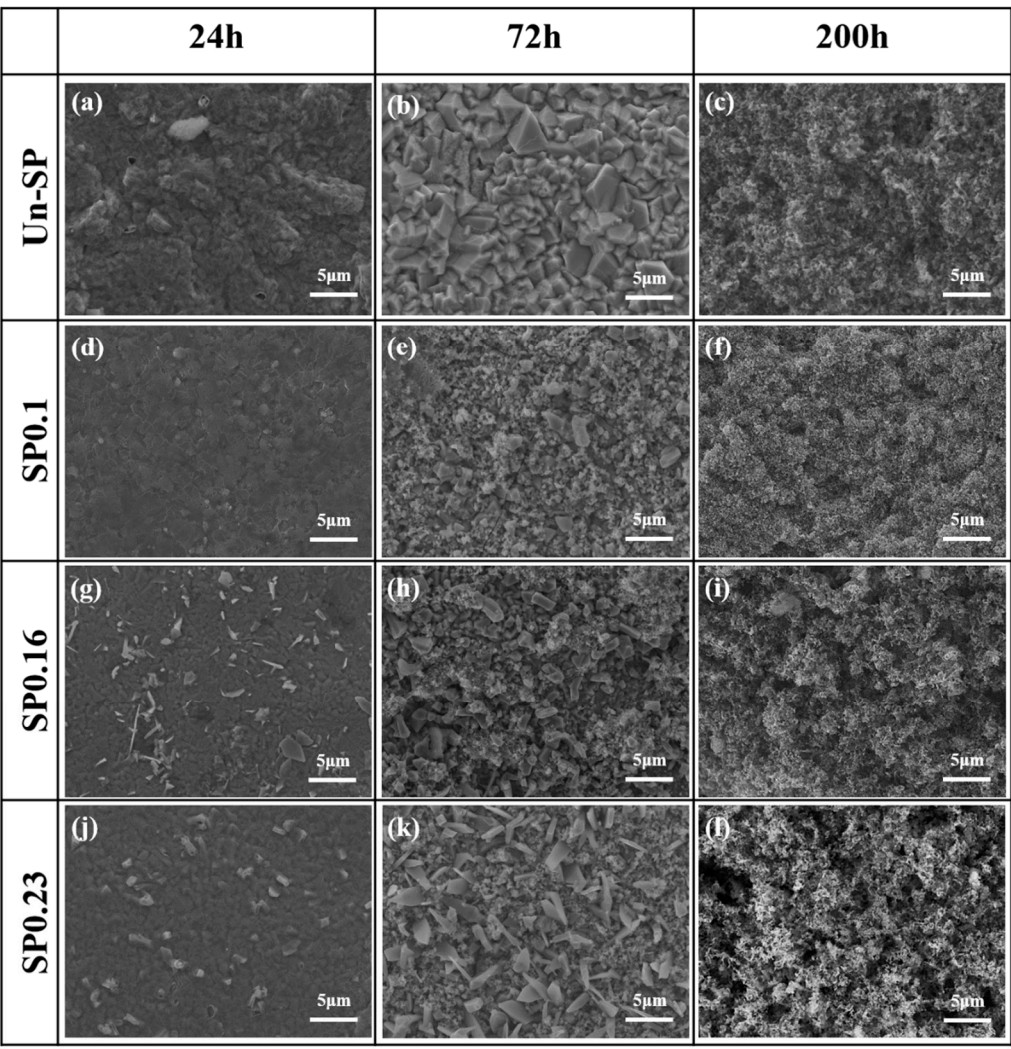

**Figure 4.** SEM surface morphologies of (**a**–**c**) Un-SP, (**d**–**f**) SP0.1, (**g**–**i**) SP0.16, and (**j**–**l**) SP0.23 samples after oxidation for 24 h, 72 h, and 200 h in 650 °C steam environment.

Figure 5 shows the XRD analysis results of the sample surface layers after steam oxidation for 72 h and 200 h. The XRD spectra indicate that after oxidation for 72 h, only

one diffraction peak of $Fe_3O_4$ appears for the surface layer of the Un-SP sample; in contrast, for the SP samples, there are diffraction peaks for both $Fe_3O_4$ and $Fe_2O_3$. These results suggest that there is a higher proportion of the $Fe_2O_3$ phase in the oxide films of the SP samples, whereas the oxide film of the Un-SP sample is primarily composed of the $Fe_3O_4$ phase, with no detectable $Fe_2O_3$ phase. After oxidation for 200 h, the diffraction peaks of both the $Fe_2O_3$ and $Fe_3O_4$ phases are present in the XRD patterns of all samples, indicating the formation of a higher proportion of the $Fe_2O_3$ phase in the oxide film.

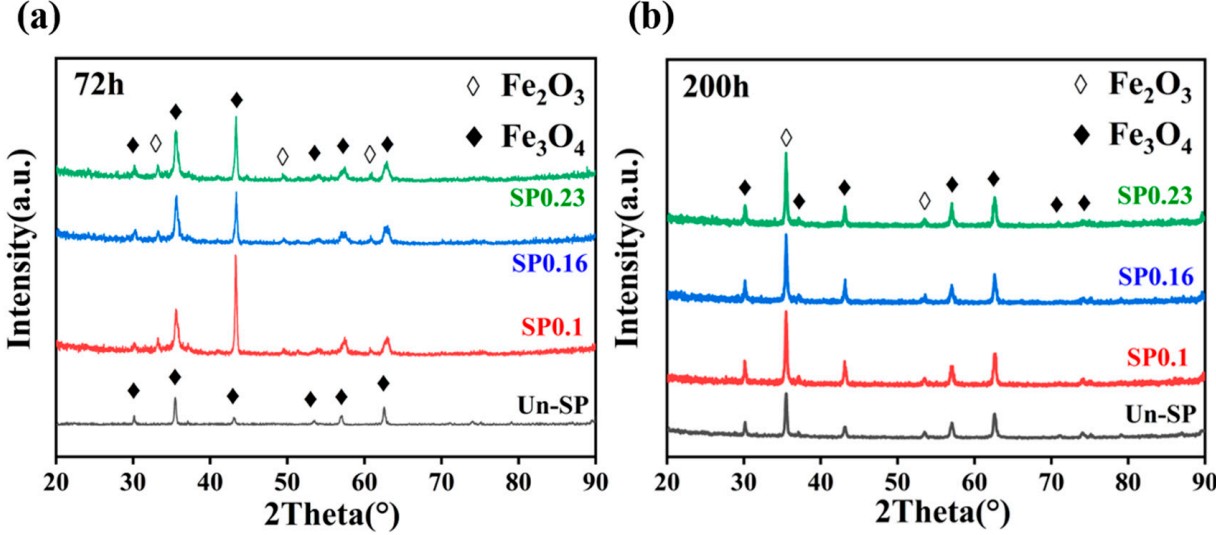

**Figure 5.** XRD patterns of Un-SP and SP samples after steam oxidation for (**a**) 72 h and (**b**) 200 h.

Figure 6 shows the high-magnification EDS point scanning images of the surfaces of Un-SP and SP0.23 in specific regions after oxidation for 72 h and 200 h, and the EDS analysis results are summarized in Table 4. After oxidation for 72 h, the surface layer of the Un-SP sample has an iron mass fraction of 72%~75%, which is equivalent to that in $Fe_3O_4$, indicating the formation of a dense particulate $Fe_3O_4$ layer. In the surface layer of the SP0.23 sample, there are white needle-shaped oxides with an iron mass fraction of 71.4%, which is close to that in $Fe_3O_4$, and the Fe to O ratio in the fluffy structures is close to that in $Fe_2O_3$. These suggest the formation of composite oxides of wrinkled $Fe_2O_3$ and particulate $Fe_3O_4$ in the surface layers of the SP samples. After oxidation for 200 h, $Fe_2O_3$ is simultaneously produced in the outer oxide layers of the Un-SP and SP0.23 samples, indicating an alternating distribution of $Fe_2O_3$ and $Fe_3O_4$. In this case, $Fe_2O_3$ is produced preferentially due to its lower Gibbs free energy requirement, probably leading to the peeling of the oxide layers due to differential thermal expansion.

**Table 4.** EDS point analysis results at the marked positions in Figure 6.

| Position | Element Content (at%) | | | | | |
|---|---|---|---|---|---|---|
| | **O** | **Cr** | **Fe** | **Mn** | **Co** | **Cu** |
| P1 | 19.02 | 3.44 | 75.42 | 0.22 | 0.46 | 0.18 |
| P2 | 23.02 | 2.46 | 72.18 | 0.45 | 0.27 | 0.53 |
| P3 | 24.61 | 1.39 | 71.41 | 1.01 | 0.03 | 0.69 |
| P4 | 29.87 | 0.42 | 66.67 | 0.56 | 0.16 | 0.67 |
| P5 | 33.02 | 1.73 | 62.13 | 0.45 | 0.27 | 1.53 |
| P6 | 31.8 | 0.37 | 64.2 | 0.49 | 0.25 | 0.96 |

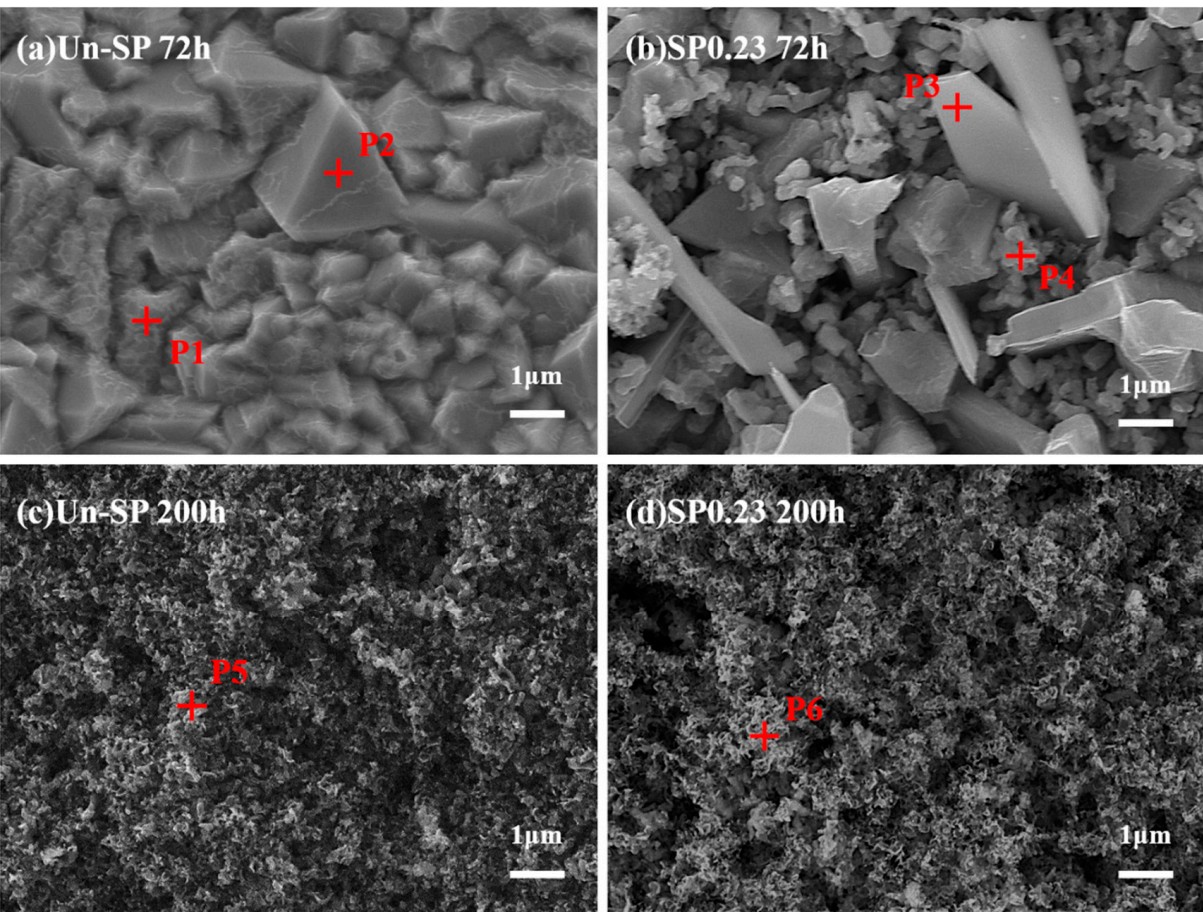

**Figure 6.** Surface morphologies of (**a**) Un-SP and (**b**) SP0.23 samples after 72 h of oxidation and (**c**) Un-SP and (**d**) SP0.23 samples after 200 h of oxidation.

### 3.4. Cross-Sectional Morphologies

Figure 7 presents the SEM cross-sectional morphologies of the Un-SP sample after oxidation at 650 °C at different times. As the oxidation process proceeds, the oxide scales on the Un-SP sample are divided into three layers with clear and smooth boundaries. There are numerous large-sized pores and gaps in the outer oxide layer, facilitating the formation of cracks. The EDS analysis results of the surface oxide scales indicate that the outer oxide layer is primarily composed of $Fe_2O_3$ and $Fe_3O_4$. During the oxidation process, a diffusion zone is segregated in the inner oxide layer of the Un-SP sample. The EDS area scanning results in Figure 8 reveal the presence of Fe, O, Cr, Co, and Mn elements in the entire region, of which the Fe content shows a notable decrease in the inner oxide layer compared to that in the outer oxide layer and the substrate. This is probably attributed to the strong affinity between Fe and O atoms, leading Fe to continuously diffuse outward and causing the growth of the outer oxide layer. As the boundary between the inner oxide layer and the diffusion zone becomes distinct, numerous black precipitates emerge at the boundary after 200 h of oxidation. These black precipitates may be precipitated phases formed by the oxidation of the alloying elements. Simultaneously, the boundary between the diffusion zone and the substrate also becomes distinct, accompanied by the formation of a few pores.

Figure 9 presents the SEM cross-sectional morphologies of SP samples oxidized at various times at different SPIs. Compared to the Un-SP sample, the SP samples have more irregular interfaces owing to the compressive stress applied by the projectiles during shot peening, which fragments the surface grains and produces an SP layer of a certain thickness, indirectly causing a fluctuation in the oxide layer interfaces. In the initial stages of oxidation, the morphology of the SP layer is observable. The EDS area scanning analysis of the SP0.23 sample after 24 h of oxidation (Figure 10a) reveals early Cr enrichment in

the outer oxide layer, probably due to the elongation effect of shot peening on the surface grains, providing pathways for Cr atoms to diffuse outward. After 200 h of oxidation, the SP layer begins to blur, showing a degrading or disappearing tendency. The disappearance of the SP layer can be attributed to the formation and continuous growth of fragmented grains, leading to the elimination of the grain boundaries. Unlike the SP0.1 and SP0.16 samples, a distinct layer between the SP layer and the substrate is observed in the SP0.23 sample. The EDS area scanning analysis of the SP0.23 sample after 200 h of oxidation (Figure 10b) shows significant Cr enrichment at this interface. The formation of the Cr-rich layer at the interface hinders the further diffusion of O into the material interior, protecting the substrate from oxidation. Simultaneously, the outward diffusion of Cr atoms in the SP layer causes a reaction with the Fe and O atoms to produce Fe-Cr oxides, which can replace the gradually degrading SP layer as the inner oxide layer.

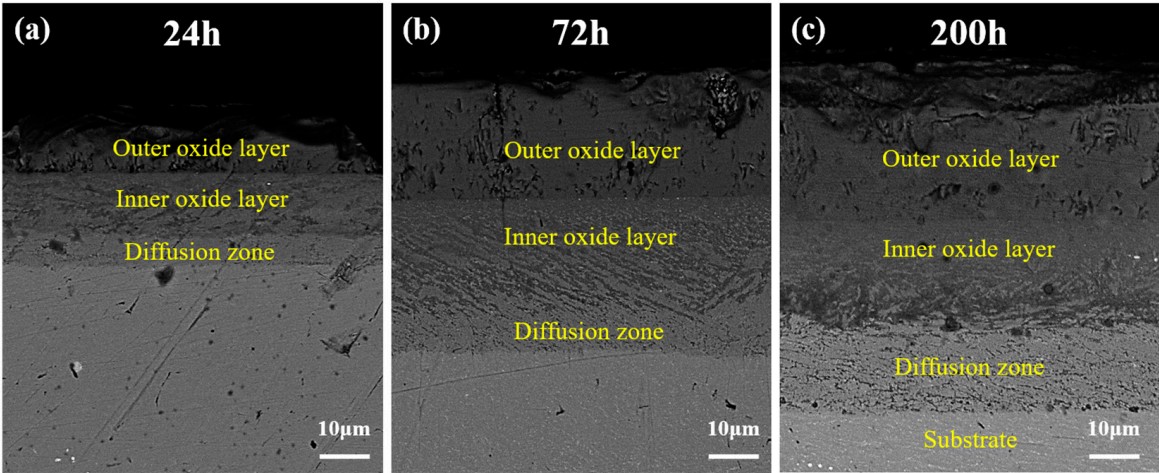

**Figure 7.** SEM-BSE cross-sectional morphologies of Un-SP sample after steam oxidation for (**a**) 24 h, (**b**) 72 h, and (**c**) 200 h.

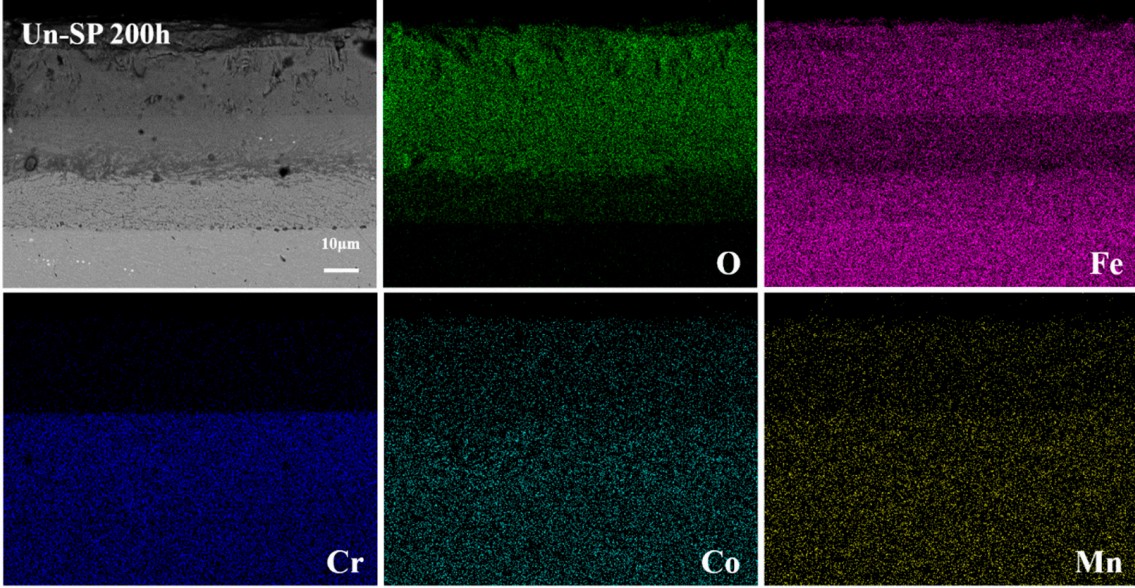

**Figure 8.** BSE image and EDS mapping results of Un-SP sample after steam oxidation for 200 h.

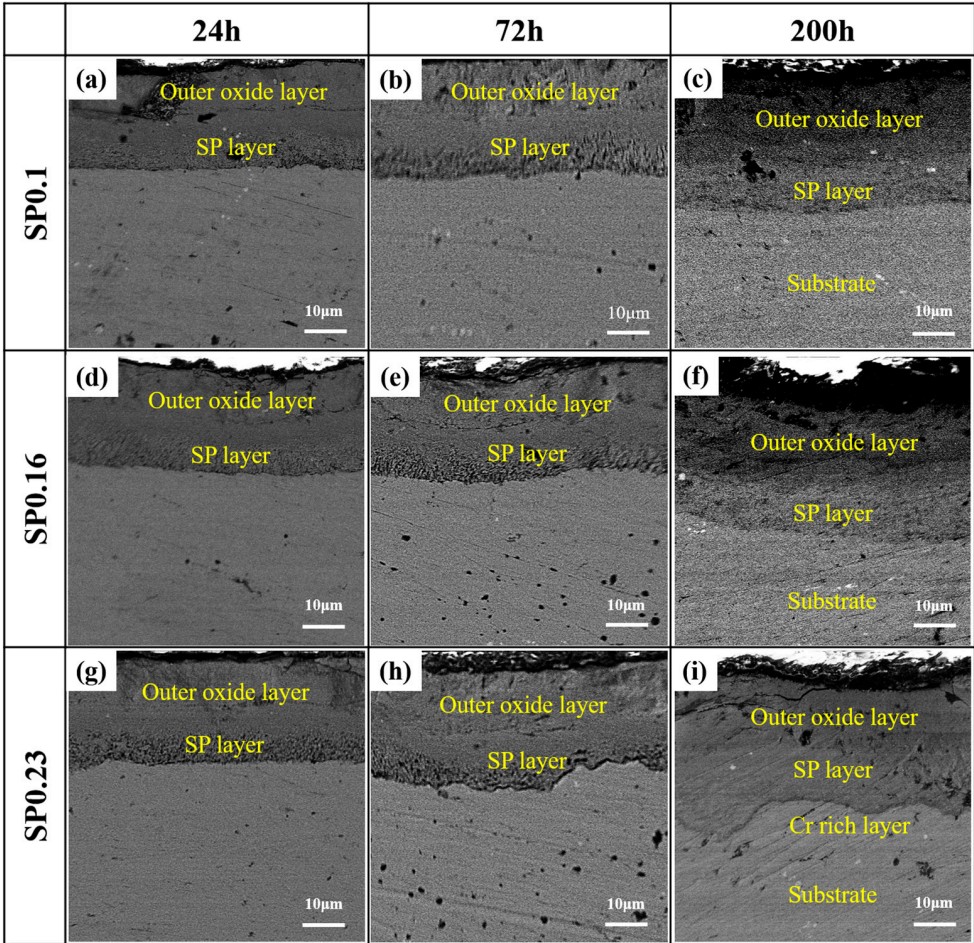

**Figure 9.** SEM-BSE cross-sectional morphologies of different SPI samples: (**a–c**) SP0.1, (**d–f**) SP0.16, and (**g–i**) SP0.23 after steam oxidation for 24 h, 72 h, and 200 h.

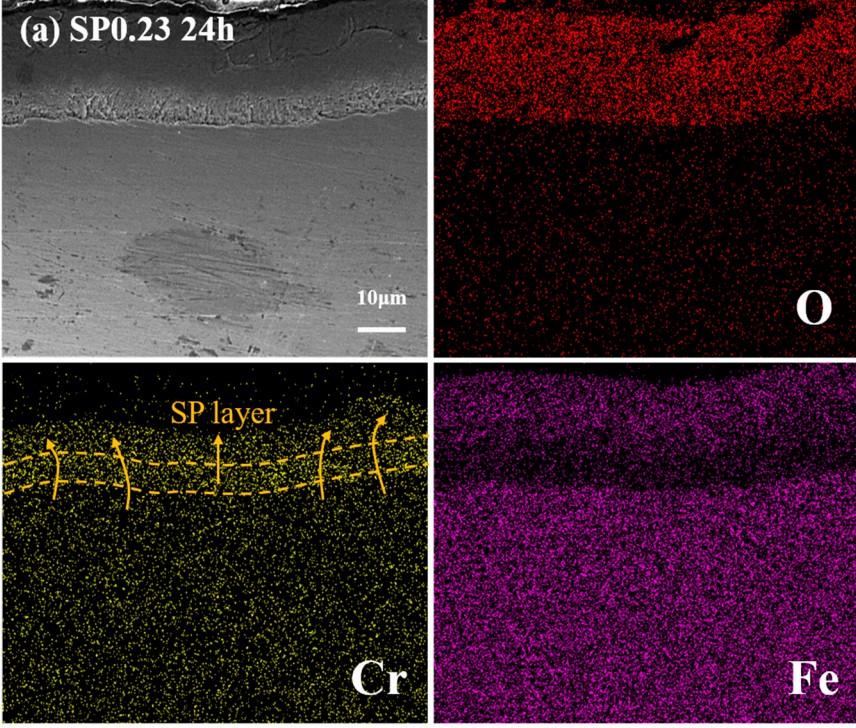

**Figure 10.** *Cont*.

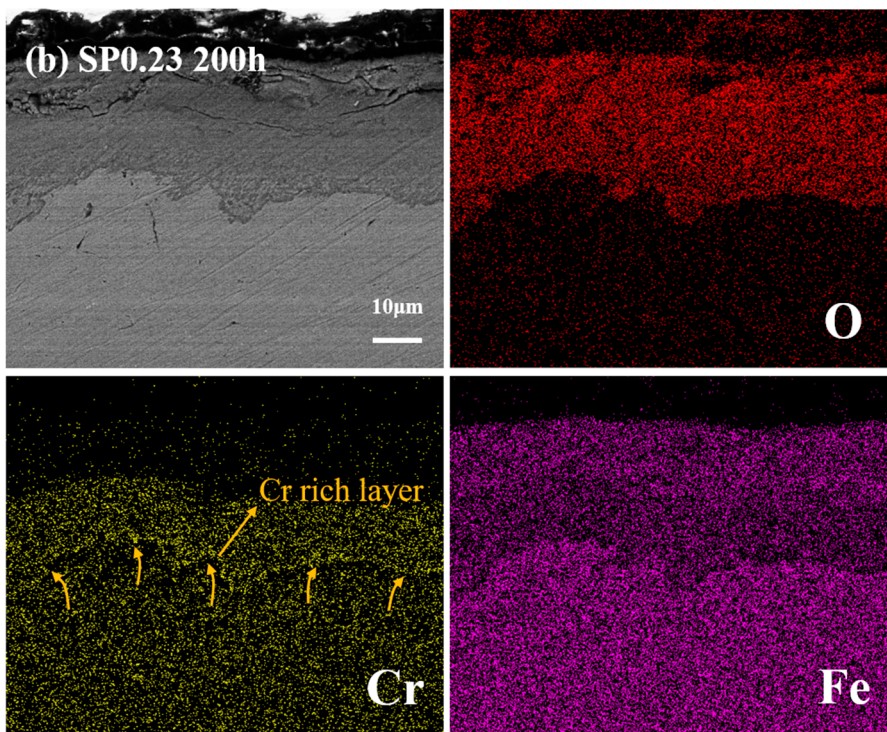

**Figure 10.** BSE image and EDS mapping results of SP0.23 sample after steam oxidation for (**a**) 24 h and (**b**) 200 h.

Figure 11 shows the TEM morphology at the interface between the SP layer and the substrate of the SP0.23 sample after steam oxidation for 200 h. A distinct boundary can be observed between the SP layer and the substrate. The oxide in the SP layer exhibits poor adhesion; it is characterized by an uneven morphology and associated peeling. In contrast, the substrate exhibits clear grains and a uniform microstructure. Further TEM analysis reveals the presence of a dense oxide layer at the interface between the SP layer and the substrate, without other metal phases or extensive pores. Through the combination of the SAED patterns (Figure 11b) and point scanning results (Figure 11c), the phase in this layer can be identified as $FeCr_2O_4$. Thus, the Cr-rich layer at the interface between the SP layer and the substrate is primarily composed of a dense $FeCr_2O_4$ oxide layer.

Figure 12 compares the thicknesses of the oxidation layer in the Un-SP and SP samples at different SPIs after oxidation for different times (24 h, 72 h, and 200 h). The thickness corresponding to each point is shown in Table 5. Throughout the oxidation process, the thicknesses of both the outer and inner oxide layers of the Un-SP samples are higher than those of the SP samples, aligning with the results of the oxidation kinetics analysis. This indicates that SP treatment can enhance the oxidation resistance of G115 steel. Among the SP samples, the SP0.23 sample shows a slightly larger thickness in the outer oxide layer after 24 h of oxidation compared to other samples. This may be due to the fact that a higher SPI is conducive to the formation of surface defects, which enhances the adsorption efficiency of water vapor and accelerates the nucleation rate of surface oxides. As a result, the formation of oxides occurs more easily at the outer layer of the SP0.23 sample. After a certain period of oxidation, the thicknesses of both the outer and inner oxide layers of the SP0.23 sample are significantly smaller than those of other samples. This could be attributed to the formation of a Cr-rich layer, which can inhibit the diffusion of O atoms into the substrate, thus hindering the oxidation behavior and slowing the increase in the oxide layer thickness.

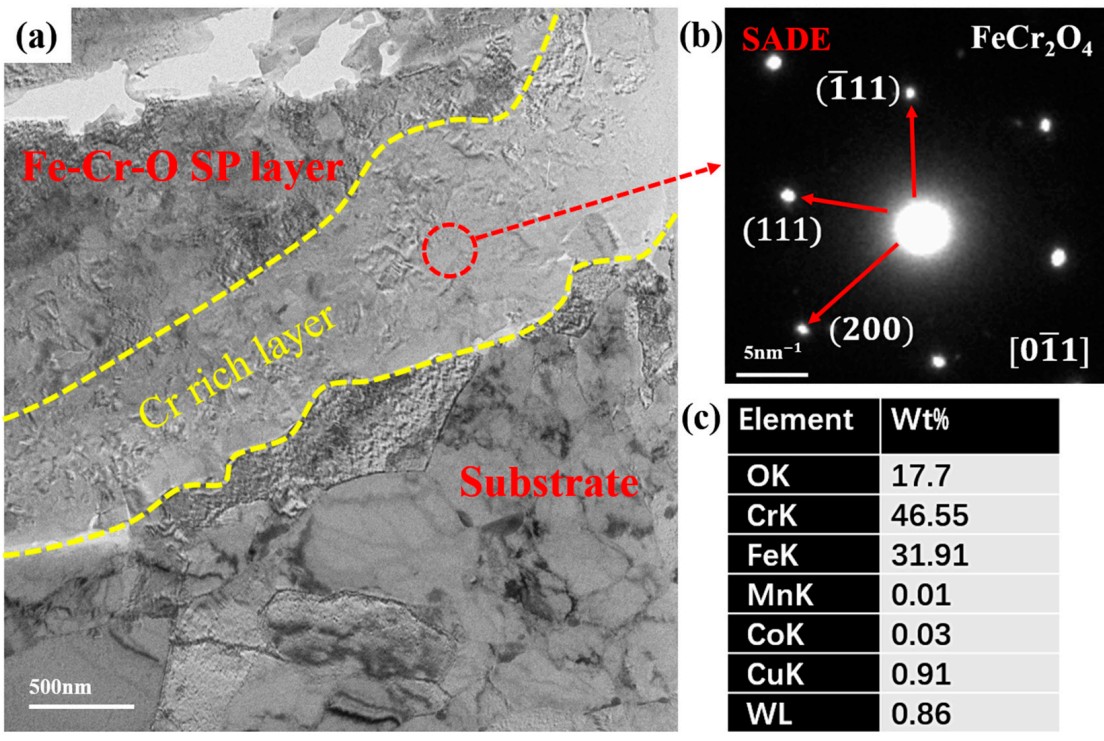

**Figure 11.** (**a**) TEM morphology of the SP0.23 sample at the interface between the SP layer and the substrate after steam oxidation for 200 h; (**b**) SAED pattern and (**c**) point scanning results of the selected area in (**a**).

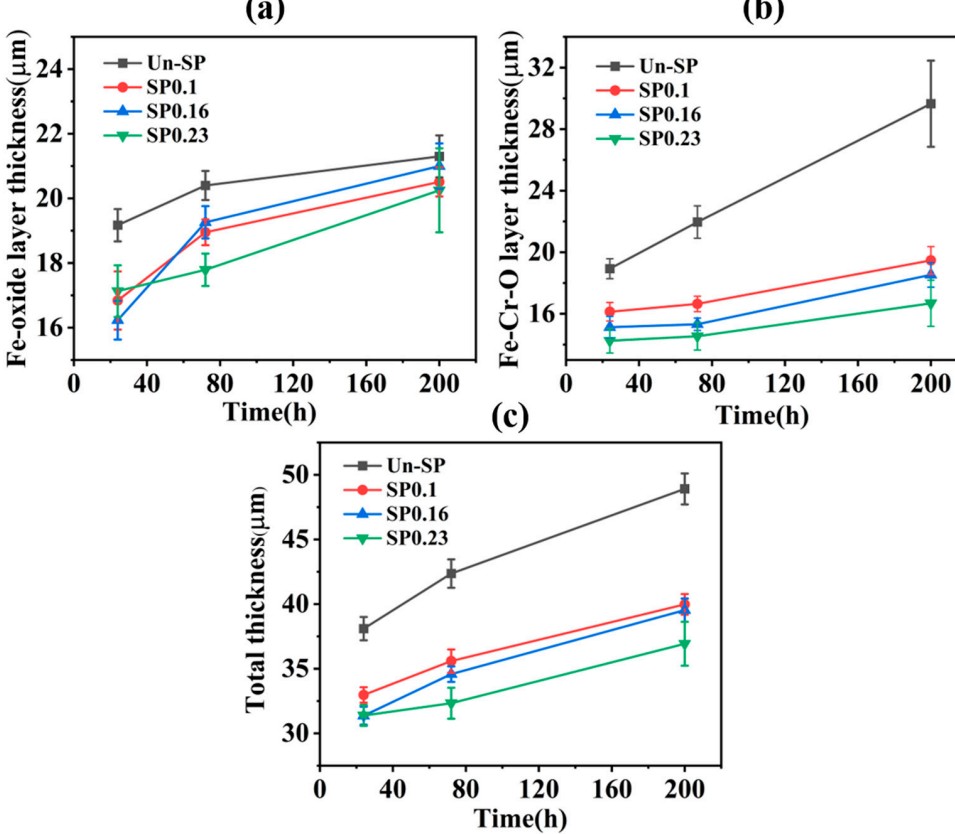

**Figure 12.** Oxide layer thickness as a function of oxidation time for Un-SP and SP samples after steam oxidation at 650 °C: (**a**) outer Fe oxide layer ($Fe_2O_3$ and $Fe_3O_4$), (**b**) SP layer (Fe-Cr-O), (**c**) total oxide layer thickness.

**Table 5.** The oxidation layer thicknesses of Un-SP and SP samples after steam oxidation at different times (Fe oxide refers to the combination of $Fe_2O_3$ and $Fe_3O_4$).

| Time | | 24 h | 72 h | 200 h |
|---|---|---|---|---|
| **Total Thickness (μm)** | Un-SP | 38.10 ± 0.90 | 42.36 ± 1.10 | 48.91 ± 1.20 |
| | SP0.1 | 32.97 ± 0.60 | 35.59 ± 0.90 | 39.98 ± 0.80 |
| | SP0.16 | 31.36 ± 0.70 | 34.58 ± 0.60 | 39.53 ± 0.90 |
| | SP0.23 | 31.38 ± 0.80 | 32.33 ± 1.20 | 36.93 ± 1.70 |
| **Fe Oxide Layer Thickness (μm)** | Un-SP | 19.17 ± 0.50 | 20.40 ± 0.45 | 21.30 ± 0.65 |
| | SP0.1 | 16.84 ± 0.90 | 18.95 ± 0.40 | 20.51 ± 0.45 |
| | SP0.16 | 16.23 ± 0.60 | 19.26 ± 0.50 | 21.00 ± 0.70 |
| | SP0.23 | 17.13 ± 0.80 | 17.79 ± 0.50 | 20.25 ± 1.30 |
| **Fe-Cr-O Layer Thickness (μm)** | Un-SP | 18.93 ± 0.65 | 21.96 ± 1.05 | 29.65 ± 2.80 |
| | SP0.1 | 16.13 ± 0.60 | 16.64 ± 0.50 | 19.47 ± 0.90 |
| | SP0.16 | 15.13 ± 0.70 | 15.32 ± 0.40 | 18.53 ± 0.80 |
| | SP0.23 | 14.25 ± 0.80 | 14.54 ± 0.90 | 16.68 ± 1.50 |

## 4. Discussion

Figure 13 presents the Ellingham diagram for the formation of Fe and Cr oxides under standard conditions. An Ellingham diagram is a graphical tool used to depict the relationship between the change in Gibbs free energy (ΔG) and the temperature for various chemical reactions [30], illustrating the thermodynamic stability of metal oxide reduction reactions. By comparing the free energy changes of different reactions at specific temperatures, it is possible to predict the priority of oxide formation for different metal elements under certain conditions. As can be seen from the figure, the Gibbs free energy order of the reaction at 650 °C is $\Delta G_{Cr2O3} < \Delta G_{Fe3O4} < \Delta G_{Fe2O3}$, which demonstrates that $Cr_2O_3$ will be generated preferentially over $Fe_3O_4$ and $Fe_2O_3$, indicating that the affinity of Cr to O is greater than Fe, and selective oxidation will preferentially take place for diffusion to the outer layer.

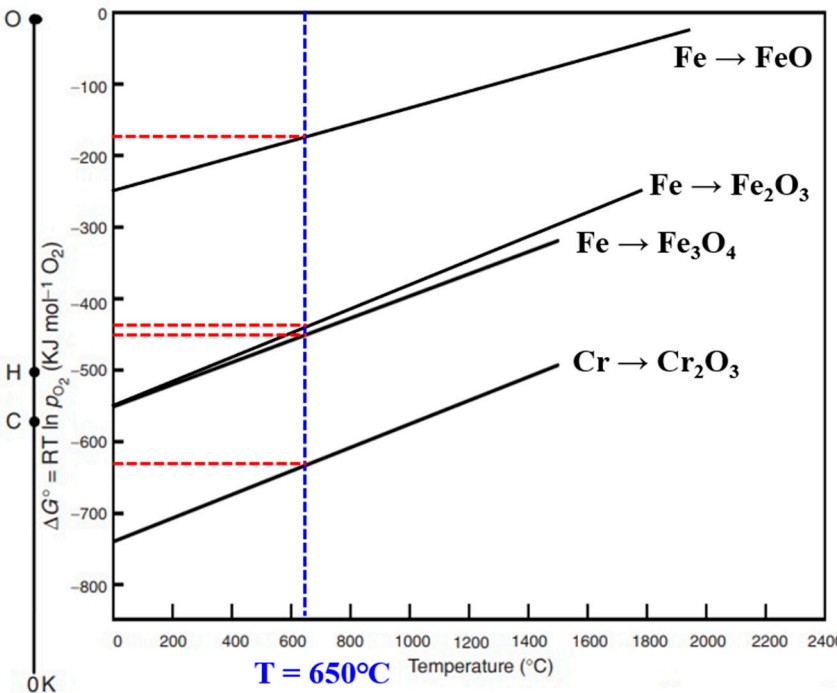

**Figure 13.** Ellingham diagram: Gibbs free energy for generation of oxides of FeO, $Fe_2O_3$, $Fe_3O_4$, and $Cr_2O_3$. The data in the figure are from [31].

Based on these results, it is possible to summarize the oxidation behavior of the SP samples in high-temperature steam environments and the mechanism by which SP enhances the oxidation resistance of G115 steel. In the initial stages of oxidation, Cr, due to selective oxidation, preferentially diffuses outward through diffusion channels produced by shot peening. Its corrosion resistance can initially hinder some of the O atoms from diffusing inward, providing the first protection. With the extension of time, the diffusion speed of the Cr atoms slows due to the low Cr content in martensitic steel, and the Fe atoms in the substrate begin to diffuse outward. They can react with the Cr and O atoms in the SP layer to form $(Fe, Cr)_xO_y$ and simultaneously diffuse to the material surface to produce an outer oxide layer mainly composed of $Fe_3O_4$. As the oxidation intensifies, since the Gibbs free energy required to form $Fe_2O_3$ is lower than that for $Fe_3O_4$, $Fe_2O_3$ preferentially nucleates and grows on the surface layer, covering $Fe_3O_4$ and producing a dual-layered structure of $Fe_2O_3$ and $Fe_3O_4$ in the outer oxide layer. Moreover, the substantial consumption of Fe in the outer layer necessitates replenishment from the interior, which leads to the massive outward diffusion of Fe from the SP layer, thus forming a Fe-depleted zone. The vacancies left by the diffused Fe atoms are then filled by Cr atoms from the substrate, and these slowly migrating Cr atoms gradually accumulate at the interface between the SP layer and the substrate, producing a dense $FeCr_2O_4$ protective layer. Compared to the G115 base material, the formation of the $FeCr_2O_4$ protective layer replaces the diffusion zone in the oxidation of the base material, significantly hindering the diffusion of O atoms into the substrate and providing further protection. Through these two protection mechanisms, the oxidation layer thickness of the SP samples is considerably reduced compared to that of the G115 base material, exhibiting a slower oxidation layer growth rate and superior oxidation resistance.

Based on the above analysis of the oxidation process of Un-SP and SP samples, the schematic diagrams of the oxidation process for Un-SP and SP samples in the 650 °C steam environment can be obtained, as summarized in Figures 14 and 15.

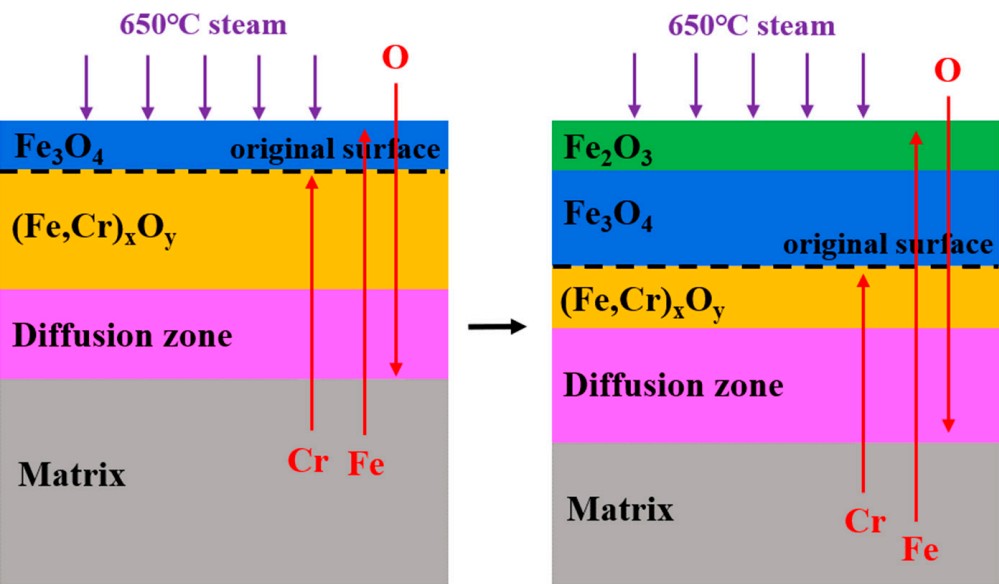

**Figure 14.** Schematic diagram of oxidation process of Un-SP G115 base material in 650 °C steam environment.

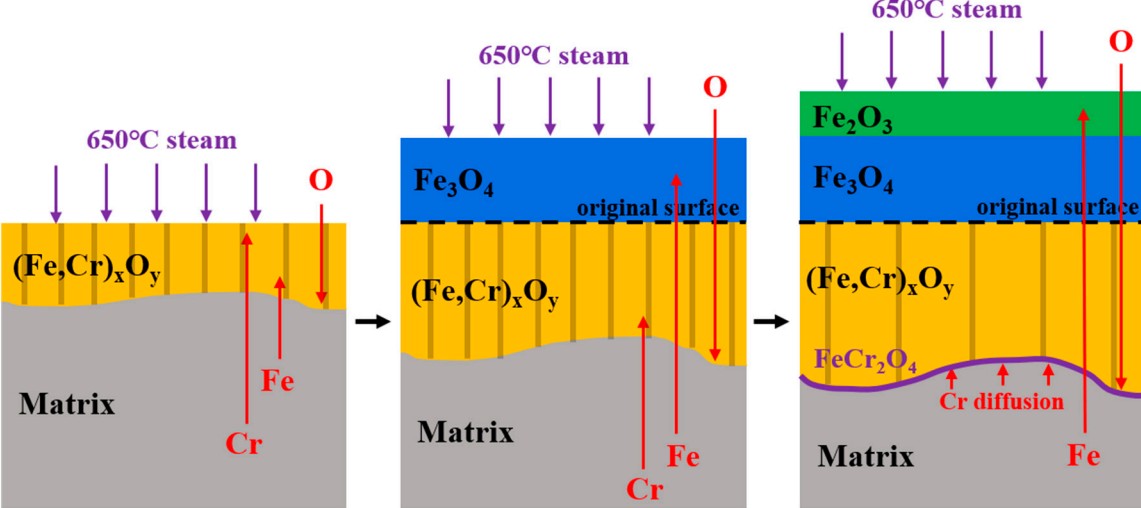

**Figure 15.** Schematic diagram of oxidation process of SP samples in 650 °C steam environment.

## 5. Conclusions

In this paper, a shot peening surface treatment method was used to refine the surface grains of martensitic heat-resistant G115 steel. The differences in the oxidation behavior of G115 steel before and after shot peening in the 650 °C steam environment were explored, and the mechanism by which shot peening improved the oxidation resistance of G115 steel was summarized. The main conclusions are as follows.

1. Shot peening leads to grain fragmentation in the surface layer of G115 steel, increasing the grain boundaries and providing abundant diffusion pathways for the selective oxidation and diffusion of Cr atoms to the surface.
2. Shot-peened samples exhibit significantly a lower oxidation weight gain compared to that of the G115 base material after oxidation in the 650 °C steam environment and a greater shot peening intensity can result in less weight gain. The oxidation of the G115 base material produces a three-layer structure comprising an outer oxide layer, an inner oxide layer, and a diffusion zone, and each layer is stable and smooth. In contrast, a dual-layer structure with outer and shot peening layers is produced on the surfaces of shot-peened samples, and the thickness of the oxidation layer is smaller than that of the G115 base material.
3. The enhanced oxidation resistance of G115 steel subjected to shot peening is manifested in two aspects. First, shot peening induces the fragmentation of surface grains, and Cr atoms can migrate to the material surface more rapidly via diffusion channels, thus producing a protective layer. Second, the substantial enrichment of Cr at the interface between the shot peening layer and the substrate produces a continuous, dense $FeCr_2O_4$ protective layer, hindering the diffusion of O atoms into the substrate. With these two aspects of the protection mechanism, shot-peened G115 steel has better oxidation resistance than the G115 base material.

**Author Contributions:** Conceptualization, P.C. and J.Z.; methodology, P.C. and L.Y.; validation, H.L.; formal analysis, P.C.; investigation, P.C. and T.D.; data curation, P.C.; writing—original draft preparation, P.C.; writing—review and editing, P.C., J.Z., L.Y. and H.L.; supervision, B.D. and Y.L. (Yuehua Liu); project administration, C.L.; funding acquisition, Y.L. (Yongchang Liu). All authors have read and agreed to the published version of the manuscript.

**Funding:** This work was financially supported by the National Key R&D Program of China (No. 2022YFB3705300), the National Natural Science Foundation of China (No. 52034004), and the Postdoctoral Fellowship Program of CPSF (No. GZB20230515).

**Institutional Review Board Statement:** Not applicable.

**Informed Consent Statement:** Not applicable.

**Data Availability Statement:** The data are contained within the article.

**Conflicts of Interest:** Authors Tianyu Du, Yuehua Liu and Baoxin Du are currently employed at Tianjin Xinyu Color Plate Co., Ltd., Tianjin, China. The contributions to this work and manuscript were made independently, without any requirement, guidance, or input from the employer. The remaining authors declare no conflicts of interest.

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
