# Peer review of "Improving the Oxidation Resistance of G115 Martensitic Heat-Resistant Steel by Surface Treatment with Shot Peening"

_coatings, doi:10.3390/coatings14050575_

Round 1

Reviewer 1 Report

Comments and Suggestions for Authors

Excellent and very complete piece of work highlighting how shot peening treatment improves the oxidation resistance of relevant steels used in steam pipes and collectors. The paper combines very detailed characterization of steels exposed to different treatments and adequate analysis and interpretation of the results. Just a very minor detail should be corrected, in Table 5, the number of significant digits should be corrected according to the uncertainty quoted.

Comments on the Quality of English Language

Moderate english revision due to several typos

Author Response

Comments 1:  in Table 5, the number of significant digits should be corrected according to the uncertainty quoted.

Response 1: After reviewing your comment, we agreed that the significant digits should match the form of the uncertainty quoted in Table 5 (Page 15). Therefore, we have updated the number of significant digits accordingly.

Comments 2:  Moderate English revision due to several typos.

Response 2: Thank you for your suggestion. We have reviewed the article and made revisions to improve the language of some questions. We believe this will make the article more understandable.

Reviewer 2 Report

Comments and Suggestions for Authors

Article

Improving the Oxidation Resistance of G115 Martensitic Heatresistant Steel by Surface Treatment with Shot Peening

Authors

Pengweng Chen, Jingwen Zhang, Liming Yu, Tianyu Du, Huijun Li, Chenxi Liu, Yongchang Liu, Yuehua Liu and Baoxin Du

G115 steel is a novel martensitic heat-resistant steel, primarily used in main steam pipelines and collectors of ultra-supercritical thermal power units. Therefore, the study of this steel to increase the service life of pipelines is important.

The dynamics of the oxidation processes of pipeline steel, the object of research, is the main factor for the longevity of the pipeline.

The research methodology is appropriate, the equipment is modern. The methodology is consistent and detailed.

Comments:

Does the small change in the mass of the surface area of the samples (10x8x2.5 mm) in the context of the accuracy of the result in the oxidation process correlate with the accuracy of determining the change in mass by weighing??

Lack of consistency in detailing analytical etc. measuring equipment - model, manufacturer, city, country)

Author Response

Comments 1: Does the small change in the mass of the surface area of the samples (10 x 8 x 2.5 mm) in the context of the accuracy of the result in the oxidation process correlate with the accuracy of determining the change in mass by weighing?

Response 1: Thank you for your question. In this experiment, the samples did not show any signs of flaking of the oxide skin. This lack of flaking resulted in a small change in surface area, which has little impact on the accuracy of determining the change in mass by weighing. It is also well-known that in vapor oxidation, the effect of surface area change on the accuracy of the oxidation weight gain curve is minimal and can be considered negligible compared to the mass change.

Comments 2: Lack of consistency in detailing analytical etc. measuring equipment - model, manufacturer, city, country)

Response 2: Thank you for bringing this to our attention. We appreciate your comments and fully agree with them. Firstly, The vapor oxidation experiments in this paper were designed based on several works of literature, including 10.1016/j.corsci.2021.110008, etc. To ensure consistency in analysis, we also used analytical characterization tools from these works. In addition, the details of the oxidation experiment are supplemented as follows.

Table. The major equipment of steam oxidation platform and its manufacturer

Equipment

Manufacturer

Model

City

Country

Tube Furnace

Hefei Kejing Material Technology Co.

OTF-1200X-S

Hefei

China

Steam Generator

Suzhou Huaxiang Shida Environmental Protection Technology Co.

HSG02

Suzhou

China

Pump

Huiyu Weiye (Beijing) Fluid Equipment Co.

BT100J-1A

Beijing

China

Reviewer 3 Report

Comments and Suggestions for Authors

1. Dear authors are encouraged to specify more clearly the aim and especially the novelty of their research. What are the specific improvements compared to well-known published works on G115 steel? Are high temperature steam and SP treatments known methods to improve oxidation resistance? If yes, what is the novelty and uniqueness of the research? How was anti-corrosion treatment of G115 steel, which is already used as ”ultra- according supercritical thermal power units”, previously carried out?

2. ….shot peening intensity (SPA)….?

3. What is the dimension of the rate constant k in eq. 2?

4. What are the dimensions of the rate constants in eqs. 3-6? Why they are different? What is physical sense of these rate constants? How is this result explained using formal chemical kinetics?

5. Three experimental points are insufficient (Fig.3).

6. Respected authors are encouraged to prove the possibility of applying classic thermodynamics for such low partial pressures of components and explain of its physical meaning (Fig. 13). Dear authors are recommended to calculate the quantity of molecules/atoms in 1 L at such low partial pressures and hence to justify using thermodynamic laws.

Comments on the Quality of English Language

English is quite understandable but it seems to be non-native

Author Response

Many thanks for your comments, please see the attachment.
